# Investigating the impact of psychedelic drugs on social cognition defects: A scoping review protocol

**Sarah Ann Smith[1], Shaina Smith[2], Liz Dennett[3], Yanbo Zhang[4]***

**1** Faculty of Science, Western University, London, Ontario, Canada, **2** Department of Biomedical and Molecular Sciences, Queen's University, Kingston, Ontario, Canada, **3** Geoffrey and Robyn Sperber Health Sciences Library, University of Alberta, Edmonton, Alberta, Canada, **4** Department of Psychiatry, Faculty of Medicine and Dentistry, University of Alberta, Edmonton, Alberta, Canada

\* yanbo9@ualberta.ca

**Data Availability Statement:** No datasets were generated or analysed during the current study. All relevant data from this study will be made available upon study completion.

## Abstract

### Background

Impairments in social cognition are known to be a key factor in several psychiatric and neurodevelopmental disorders. Interest in psychedelic drugs has increased in recent years, with significant research identifying psychedelic and hallucinogenic drugs as modulators of social cognition. However, more research is necessary before psychedelics are implemented in clinical settings as treatments for social cognition defects. Therefore, this study describes a scoping review protocol which will be used to analyze the body of literature on psychedelic drugs as modulators of social cognition in patients with psychiatric and neurodevelopmental disorders.

### Methods

This scoping review protocol was developed using the JBI Scoping Review Methodology Group's description of how to conduct a scoping review. The guidelines identified by this group as well as a search strategy developed with the assistance of a research librarian will be applied to a search of several peer-reviewed journals, including MEDLINE (Ovid), PscINFO, EMBASE (Elsevier), and Scopus (Elsevier). Each study extracted will be screened in a two-step screening process, including a title and abstract screen, and a full-text screen. One independent individual will complete both steps of the screening, and a second independent individual will review the completed screening.

### Discussion

An understanding of the current literature on psychedelic drugs as modulators of social cognition will provide insight into what is presently known on the subject, and any gaps in the literature that can be addressed in future studies. The knowledge gained from this scoping review could lead to a new treatment for social cognition defects in clinical populations.

**Funding:** "This research is funded by University of Alberta Start-up Fund, University of Alberta, Canada (Grant No. RES0052505).".

**Competing interests:** The authors have declared that no competing interests exists.

## Introduction

### Background

Human beings are inherently social. We have evolved to live in communities and form social connections, being social with others is a natural and necessary part of our lives [1]. The ability to function in social situations is integral to some of the most important parts of human lives, such as maintaining a job and having a family. The skills required to interact socially are categorized under the term 'social cognition', defined as "cognition in which people perceive, think about, interpret, categorize and judge their own social behaviors and those of others" [2]. Impairments in social cognition are considered to be a key feature of several psychiatric and neurodevelopmental disorders, including schizophrenia, depression and autism [3]. Additionally, defects in social cognition are known to increase the risk of developing a psychiatric disorder and play a significant role in the maintenance of a disorder by preventing an individual from seeking help from a support system and preventing re-integration activities such as returning to work [4, 5]. The Research Domain Criteria (RDoC) initiative has included social cognition as one of six domains of human functioning, further breaking down social cognition into four constructs: affiliation and attachment, social communication, perception and understanding of self and perception and understanding of others [6].

Psychedelic drugs, also commonly known as hallucinogens, are psychoactive substances that create changes in consciousness, including changes in mood, cognitive processes and perception [7, 8]. 3-4- methylenedioxymethamphetamine (MDMA) has been shown to cause an enhanced mood, emotional warmth, empathy and a willingness to discuss emotionally charged memories [9]. Lysergic acid diethylamide (LSD) causes a feeling of closeness to others, openness, and enhanced emotional empathy [10]. Studies reveal that psilocybin enhances feelings of connectedness, increases openness, and increases altruism [11]. N, N-dimethyltryptamine (DMT) is known to create distortions in auditory and time perception, and feelings of ego dissolution [12]. Several psychedelic drugs also have dose-effect outcomes, with high doses ($\geq$ 3mg/kg) of MDMA producing antidepressant effects, some addictive potential and amnesia of fear conditioning memory, and low doses ($\leq$ 1 mg/kg) not producing any of these effects [13]. Similarly, 5 mcg of LSD has been demonstrated to be the minimal dose required for observable effects, however, 20 mcg produces the most apparent effects [14]. Interest in psychedelic drugs as psychiatric treatments has rapidly increased in recent years, including extensive literature indicating that psychedelic drugs such as 3-4-methylenedioxymethamphetamine (MDMA) and psilocybin could be possible treatments for treatment-resistant depression and post-traumatic stress disorder (PTSD) [15, 16]. Significant amounts of research have identified psychedelic drugs as modulators of social cognition, increasing compassion, and closeness, and improving interpersonal relationships among other factors [17]. Rigorous scientific studies have investigated the applicability of this function for social cognition defects in psychiatric and developmental disorders. A study conducted by Reed et al. (2019) found a normalization of emotional processing in patients with major depressive disorder (MDD) after ketamine treatment [18]. An additional clinical study investigating autistic adults found a reduction in social anxiety after treatment with MDMA [19]. A study investigating patients with MDD or bipolar depression found a ketamine-based treatment significantly improved both subjective and objective measures of social functioning [20].

### Rationale

After the COVID-19 pandemic, the rates of psychiatric and neurodevelopmental disorders (e.g. autism) have increased, making the need for effective treatments needed more than ever

[21, 22]. Additionally, while many treatments for psychiatric and neurodevelopmental disorders exist, few treatments target the social cognition defects that are involved in many mental illnesses; psychedelic drugs have the potential to fill this gap. Furthermore, there have been scoping and systematic reviews investigating the therapeutic potential of psychedelics on social cognition defects. However, most have focused on only one psychedelic drug and one psychiatric or neurodevelopmental disorder. To our knowledge, this will be the first study that will include a wide array of psychedelic drug treatments and investigate social cognition defects in several psychiatric and neurodevelopmental disorders.

## Objective

The objective of this scoping review is to determine how psychedelic drugs can act as modulators of social cognition in patients experiencing a deficit in social cognition due to a psychiatric or neurodevelopmental disorder. This scoping review also aims to identify any gaps in the literature and provide suggestions for future research that will inform the clinical use of psychedelics. Specifically, our objectives for this study include answering one primary question and four sub-questions.

I. How can psychedelics act as modulators of social cognition in patients experiencing a deficit in social cognition due to a psychiatric or neurodevelopmental disorder?

i. How do psychedelics affect social cognition on a neurobiological level?

ii. How are the effects of psychedelics on social cognition realized in patient experiences and behaviour?

iii. How have psychedelics affected social cognition in different psychiatric and neurodevelopmental disorders?

iv. How would psychedelics best be applied to clinical settings?

## Methods

We will be using a scoping review methodology to measure the impact of psychedelics on several measures of social cognition deficits as defined by research domain criteria (RDoC) (system of social processes constructs) [6]. This scoping review will be developed using the methodological framework put forth by the Joanna Briggs Institute [23]. This protocol was written based on the Preferred Reporting Items for Systematic Reviews and Meta-Analyses Protocols (PRISMA-P) Checklist (2015) guidelines and a filled checklist is available for viewing in S1 Checklist.

### Inclusion and exclusion criteria

The term 'studies' will be used to refer to literature published in peer-reviewed journals. Studies will not be excluded based on publication date. All inclusion and exclusion criteria are included in Table 1.

We will be including clinical and case studies that investigate individuals of any age, sex or ethnicity experiencing deficits in social cognition due to psychiatric illness or neurodevelopmental disorder. These disorders include autism, bipolar disorder, borderline personality disorder (BPD), depression, post-traumatic stress disorder (PTSD), and schizophrenia [24].

**Table 1. Inclusion and exclusion criteria.**

| Inclusion criteria | 1. Primarily report a modulation of social cognition by psychedelic and hallucinogenic drugs |
| --- | --- |
| | 2. Clinical studies and case studies |
| | 3. Participants are diagnosed with a psychiatric or neurodevelopmental disorder |
| | 4. Treatment includes psychedelic and hallucinogenic drugs |
| Exclusion criteria | 1. Not written in or translated to English |
| | 2. Did not investigate a modulation of social cognition by psychedelic or hallucinogenic drugs |
| | 3. Book chapters, letters, newspaper articles, review articles, conference abstracts, animal studies, *in vitro* studies |
| | 4. Could not access the full article |
| | 5. Any texts not published in a peer-reviewed journal |

Studies investigating psychedelic and hallucinogenic drugs as modulators of social cognition will be included. The drugs investigated will include 3,4-methylenedioxyamphetamine (MDMA), lysergic acid diethylamide (LSD), psilocybin, N, N-dimethyltryptamine (DMT), gamma-hydroxybutyrate (GHB), and ayahuasca.

## Search strategy

A search of the literature will be completed using the electronic databases PubMed/MEDLINE (Ovid), PsycINFO, EMBASE (Elsevier), and Scopus (Elsevier). The Cochrane Library will be searched for published clinical trials and clinical trial registry records. A search strategy will be developed for each database selected in consultation with a librarian. A search strategy for PubMed/MEDLINE (Ovid) is available for viewing in S1 File.

A limited search of the databases MEDLINE has been performed using keywords. The text words contained in the title and abstracts of retrieved papers were analyzed in addition to the index terms used to describe the articles. These terms will be used to develop a search strategy with the assistance of an experienced research librarian. This initial search was completed on February 13, 2024.

This search will be followed by a second search across all selected databases using keywords and index terms identified in the initial search. The last search conducted will include a search of the reference lists of all sources selected for full-text review.

We will only search for sources in the English language. The search strategy will include literature from the databases' inception to the present.

## Record management

All identified references will be imported into the review management software Covidence, which will remove duplicates prior to the title and abstract screening phase. Following the two-step screening phase, all data will be stored in Microsoft Word.

## Screening strategy

We will select studies in a two-step screening process, title and abstract screening, and a full-text review. One independent reviewer (SAS) will complete both steps of the screening process and a second independent reviewer (SS) will verify the completed screening to ensure all approved studies meet the inclusion and exclusion criteria. Any disagreements will be discussed between the reviewers. A third reviewer (YZ) will be consulted if a decision cannot be reached.

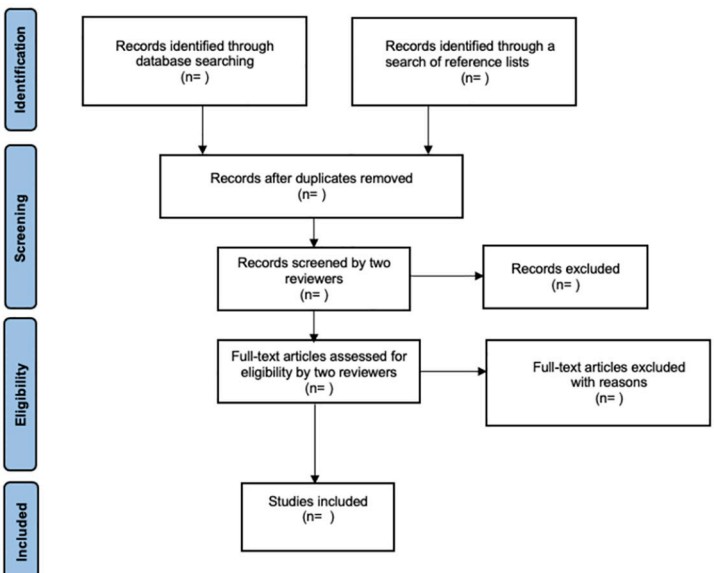

**Fig 1. Flow diagram of study selection process as described by preferred reporting items for systematic reviews and meta-analyses guidelines [25].**

Screening will be completed using the review management software Covidence. In the title and abstract screening phase, all inclusion criteria must be met and none of the exclusion criteria must be met for progression to the full-text screening phase. Following the title and abstract screening, the full texts of approved studies will be retrieved, and citation details will be imported. In the full-text screening phase, all approved studies must meet all inclusion criteria and contain none of the exclusion criteria.

Studies excluded at the full-text review will be mentioned in an excluded studies table, where reasons for exclusion will be stated. This process follows the Preferred Reporting Items for Systematic Reviews and Meta-Analyses (PRISMA) guidelines, depicted in Fig 1 [25]. Two independent reviewers (SAS and SS) will confirm the final list of included studies.

## Data extraction

Relevant data will be extracted from all included studies by one independent reviewer (SAS), while another independent reviewer (SS) checks the data extraction to ensure all relevant data is collected accurately. Data will be recorded in a structured data recording table developed by the reviewers. The extracted data will include details on author, year of publication, study design, study population, measurement scale, aims, treatment administered, dosage of treatment, duration of administration of treatment, domain of social cognition investigated, outcomes, and key findings. A draft of the data extraction form can be viewed in S1 Form. The tool will be piloted and edited accordingly by reviewers during the initial stages of data extraction. Any disagreements between reviewers will be resolved by discussion and a third reviewer (YZ) will be consulted if a decision cannot be reached. Authors of included studies will be contacted for any missing or incomplete data.

## Data summarization and presentation

To present the results, data extracted from included studies will be presented in a tabular form. The charted results will include data on the target population, treatment, dose of treatment

administered, duration of administration of treatment and the target domain of social cognition. Results will also be presented in a narrative summary, elaborating on data presented in the chart in accordance with the aims of this scoping review. At this stage, risks of bias will be avoided by highlighting discrepancies that result from population, administration, or dose-related differences. Gap identification will identify areas in the literature where there is a lack of data on psychedelic drugs and their modulation of social cognition in psychiatric and neurodevelopmental disorders.

### Stakeholder consultation

A stakeholder will be consulted to review the findings identified in the review. Stakeholders will include researchers with extensive experience investigating psychedelic drugs and novel treatments for psychiatric and neurodevelopmental disorders. Stakeholders will be presented with our scoping review, and we will collect their feedback. Their feedback will be considered upon the creation of our final report.

### Discussion

This study is a scoping review therefore no risk of bias will be carried out. Furthermore, as the reviewers only speak English, they are limited to literature published in English. Future dissemination of any findings will include publishing a full scoping review in a peer-reviewed journal. Complete searches and data collection have not commenced. No ethics approval is required for this study.

Our scoping review will aim to provide insight into the efficacy of psychedelic drugs in the modulation of social cognition in psychiatric or neurodevelopmental patients experiencing a deficit in one or multiple social cognition domains. The insights that this study will provide could possibly lead to a new treatment for social cognition deficits for use in clinical settings. We will also identify any gaps in the literature investigating the effect of psychedelic drugs on social cognition, providing direction for future research.

In this review, we aim to identify the mechanisms by which psychedelics affect social cognition on a neurobiological level. By identifying these mechanisms, we hope to guide future research by identifying possible targets for drugs that aim to ameliorate deficits in social cognition. Our scoping review will also examine evidence that provides a better understanding of how the use of psychedelic drugs impacts a patient's experiences and behaviors. This will include a reduction of social cognition defects, behavioral changes, the apparent improvement in social relationships, and overall impacts on quality of life. Including clinical evidence of effects on social cognition defects in a variety of psychiatric and neurodevelopmental disorders will also contribute to giving a complete view of the utility of psychedelic drugs in everyday clinical settings.

### Supporting information

**S1 Checklist. Preferred reporting items for systematic review and meta-analyses protocols (PRISMA-P) checklist.**
(DOCX)

**S1 Data. Data extraction form.**
(PDF)

**S1 File. MEDLINE (Ovid) search string.**
(DOCX)

## Author Contributions

**Conceptualization:** Sarah Ann Smith, Liz Dennett, Yanbo Zhang.

**Investigation:** Sarah Ann Smith.

**Methodology:** Sarah Ann Smith, Liz Dennett.

**Project administration:** Yanbo Zhang.

**Supervision:** Yanbo Zhang.

**Writing – original draft:** Sarah Ann Smith.

**Writing – review & editing:** Sarah Ann Smith, Shaina Smith, Liz Dennett, Yanbo Zhang.

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
