## [Decision Letter · Decision Letter 0]

23 May 2024

PONE-D-24-10240Investigating the impact of psychedelic drugs on social cognition defects: A scoping review protocolPLOS ONE

Dear Dr. Zhang,

Thank you for submitting your manuscript to PLOS ONE. After careful consideration, we feel that it has merit but does not fully meet PLOS ONE’s publication criteria as it currently stands. Therefore, we invite you to submit a revised version of the manuscript that addresses the points raised during the review process. Please see below the supportive comments made by Reviewer One and myself (Reviewer Two). I encourage you to consider each of the excellent points made by Reviewer One. In addition, I also ask that you respond to each of our suggestions. If there is an modification to your initial submitted *Proposal *for your Scoping Review, feel free to include a revised version in your re-submission. A re-submission of your *Proposal *will ***not*
**go back out for a second Review, but I will take a quick look at any updates you provide. Keep in mind that your Scoping Review in final form will be peer-reviewed once completed. I look forward to seeing your continued efforts towards this important area of medical science.

We look forward to receiving your revised manuscript.

Kind regards,

Herb Covington, Ph.D.

Academic Editor

PLOS ONE

Reviewers' comments:

Reviewer's Responses to Questions

**Comments to the Author**

1. Does the manuscript provide a valid rationale for the proposed study, with clearly identified and justified research questions?

Reviewer #1: Yes

Reviewer #2: Yes

2. Is the protocol technically sound and planned in a manner that will lead to a meaningful outcome and allow testing the stated hypotheses?

Reviewer #1: Yes

Reviewer #2: Yes

3. Is the methodology feasible and described in sufficient detail to allow the work to be replicable?

Reviewer #1: Yes

Reviewer #2: Yes

4. Have the authors described where all data underlying the findings will be made available when the study is complete?

Reviewer #1: Yes

Reviewer #2: Yes

5. Is the manuscript presented in an intelligible fashion and written in standard English?

Reviewer #1: Yes

Reviewer #2: Yes

6. Review Comments to the Author

You may also provide optional suggestions and comments to authors that they might find helpful in planning their study.

Reviewer #1: Review of:

Investigating the impact of psychedelic drugs on social cognition defects: A scoping review

protocol

Sarah Ann Smith, Shaina Smith, Liz Dennett, and Yanbo Zhang

Smith et al. propose a way to investigate the research related to the use of psychedelics to improve social cognition skills. This is an extremely important aspect of psychedelic treatment and deserves the wide overview that they offer to present a summary of the literature. The scoping review protocol that they propose, while limited to English language publications, focuses on social cognition defects in subjects with psychiatric diagnoses and neurodevelopmental disorders. Hopefully this will be a beginning protocol that can reveal the need for further research and possibilities for the role of psychedelics in mental health care. Furthermore, focused research will be necessary to hone the possibilities for a myriad of psychedelic substances, which vary tremendously in their effects. Smith et al. make no distinction between the different substances, or the nonclinical contexts in which they are used, highlighting the need for research that specifies set and setting, history, traditional uses and context for each substance. Each of these factors and several others greatly affect the consequences of psychedelic use. Other factors which must be queried include the prior experience of subjects with psychedelic experiences, their mental health care and context (including prior care, family history and aftercare) as well as their spiritual beliefs and experiences, and the community/cultural context within which the substances are used. While Smith et al on the one hand, throw a wide net (including a vast array of psychedelics in their protocol), on the other hand they limit their protocol to studies with subjects who have defects in social functioning. This narrow focus ignores the enormous body of research that examines psychedelic experiences for wellness, enhancing mental health for those who are seeking spiritual expansion, and the ways psychedelic experiences are tools for enhanced social cognition and social cohesion outside of a clinical setting. However, if the protocol remains narrow it can illuminate gaps in research and a possible direction for future research and funding.

Reviewer #2: The proposal to prepare a scoping review that systematically illustrates the impact of psychedelic drugs on social cognition by Smith, Smith, Dennett, and Zhang has been carefully developed and will ultimately serve as a timely contribution within a rapidly emerging clinical & experimental space. I have no doubts that - in its final form - this review will provide an informative reference for a wide audience – in line with diverse readership at PLoS One.

I very much appreciate the authors’ outlining of their PRISMA-ScR checklist. In addition, the question being asked is clear, the strategy to identify and include targeted literature appears unbiased and sound, and there is a consensus towards the plan to explore and report on the peer-reviewed & published data.

I have only three points that may increase the clarity and confidence of this Scoping Review’s message:

First, I encourage the authors to emphasize effects of specific drugs in this class, and to include precise details about dose-effect outcomes; as the pharmacology of these drugs is often biphasic -with ascending and descending limbs on behavioral, cognitive, and emotional endpoints.

Second, it will be useful to be sure an overview (perhaps additional Heading) of how any risks of bias are clearly avoided in the interpretation & consensus of meaning, as interpreted from study results. For example, I suspect highlighting discrepancies resulting from dose-related differences may bolster awareness of attention to crucial details in such a way as to avoid bias.

Finally, even though the Methods - under the Heading: Screening Strategy - mentions the use of two reviewers to verify inclusion and exclusion criteria, and even suggests a third, if necessary, I could argue there is utility in again identifying the use of two (or three reviewers) within the flow chart provided in the flow of Figure 1 (S1 Checklist).

7. PLOS authors have the option to publish the peer review history of their article (what does this mean?). If published, this will include your full peer review and any attached files.

Reviewer #1: No

Reviewer #2: **Yes: **Herbert Covington

---

## [Author Response · Author response to Decision Letter 0]

25 Jun 2024

Manuscript number: PONE-D-24-10240

Title: Investigating the impact of psychedelic drugs on social cognition defects: A scoping review protocol

Reviewer 1: 

Comment: Smith et al. propose a way to investigate the research related to the use of psychedelics to improve social cognition skills. This is an extremely important aspect of psychedelic treatment and deserves the wide overview that they offer to present a summary of the literature.

Author’s Response and Changes: Thank you for your feedback.

Comment: The scoping review protocol that they propose, while limited to English language publications, focuses on social cognition defects in subjects with psychiatric diagnoses and neurodevelopmental disorders. Hopefully this will be a beginning protocol that can reveal the need for further research and possibilities for the role of psychedelics in mental health care. Furthermore, focused research will be necessary to hone the possibilities for a myriad of psychedelic substances, which vary tremendously in their effects.

Author’s Responses and Changes: Thank you for your feedback.

Comment: Smith et al. make no distinction between the different substances or the nonclinical contexts in which they are used, highlighting the need for research that specifies the set and setting, history, traditional uses and context for each substance. Each of these factors and several others greatly affect the consequences of psychedelic use. Other factors which must be queried include the prior experience of subjects with psychedelic experiences, their mental health care and context (including prior care, family history and aftercare) as well as their spiritual beliefs and experiences, and the community/cultural context within which the substances are used.

Author’s Response and Changes: Thank you for outlining the key factors that may affect users' perspectives and experiences of psychedelics, which will further impact social cognition. In our scoping review, addressing the limitation of not distinguishing between different substances and nonclinical contexts in psychedelic research requires a multifaceted approach:

Thank you for outlining the key factors that may affect users' perspectives and experiences of psychedelics, which will further impact social cognition. In our scoping review, addressing the limitation of not distinguishing between different substances and nonclinical contexts in psychedelic research requires a multifaceted approach:

The authors will pay attention to the set and setting in which each psychedelic drug is administered in included studies, taking note of factors such as the participant's mindset, expectations, and the physical environment where the experience takes place. Any provided information will be included in the scoping review. Furthermore, research will also be performed on the nonclinical contexts in which they are used 

The authors will also conduct extensive research on each substance's historical and traditional context and include this information in the scoping review. Future clinical studies should, therefore, consider such information when designing their studies.

The authors will note any information provided on participants' prior experience with psychedelic drugs, mental health history, ongoing mental health care, spiritual beliefs, and cultural background and provide the information when discussing each included study. If information is not provided, it will be noted in the scoping review and mentioned that further studies should investigate these factors.

The scoping review will note any aftercare protocols followed in the included studies and stress their importance for maximizing benefits and minimizing adverse effects. Furthermore, any longitudinal studies that demonstrate the long-term effects and outcomes of psychedelic treatments will be discussed. If there is a lack of longitudinal studies, this will be noted, and the importance of such studies will be stressed.

Moreover, the importance of incorporating perspectives from psychology, neuroscience, anthropology and cultural studies in the investigation of psychedelics will be mentioned as a diverse perspective on psychedelic drugs is critical to understanding the many aspects that impact the usage and benefits of psychedelic drugs.

Comment: While Smith et al on the one hand, throw a wide net (including a vast array of psychedelics in their protocol), on the other hand they limit their protocol to studies with subjects who have defects in social functioning. This narrow focus ignores the enormous body of research that examines psychedelic experiences for wellness, enhancing mental health for those who are seeking spiritual expansion, and the ways psychedelic experiences are tools for enhanced social cognition and social cohesion outside of a clinical setting. However, if the protocol remains narrow it can illuminate gaps in research and a possible direction for future research and funding.

Author’s Response and Changes: We thank the reviewer for this insightful observation and comments. It is a hard balance to include all possible areas and multiple psychedelics in one scoping review. On the one hand, we are aware of the widespread impact psychedelics can have on the human body, mind and spirit; these findings are fascinating and critical for understanding the biopsychosocial influence of psychedelics; on the other hand, handling such a comprehensive topic is a challenging task and may easily dilute the focus and lose the target audience. Thus, we chose to structure our protocol to be useful to a certain audience (e.g., clinicians) and narrowed the social cognition domain that is mostly relevant to clinicians’ daily tasks. Furthermore, narrowing the protocol enables us to identify any positive or negative effects of psychedelics and gaps in the literature that would be specifically relevant to the clinical population, allowing our study to be the most beneficial it possibly can be to clinicians who work with patients with social cognition deficits.

Reviewer 2: 

Comment: The proposal to prepare a scoping review that systematically illustrates the impact of psychedelic drugs on social cognition by Smith, Smith, Dennett, and Zhang has been carefully developed and will ultimately serve as a timely contribution within a rapidly emerging clinical & experimental space. I have no doubts that - in its final form - this review will provide an informative reference for a wide audience – in line with diverse readership at PLoS One.

Author’s Response and Changes: Thank you for your feedback

Comment: I very much appreciate the authors’ outlining of their PRISMA-ScR checklist. In addition, the question being asked is clear, the strategy to identify and include targeted literature appears unbiased and sound, and there is a consensus towards the plan to explore and report on the peer-reviewed & published data.

I have only three points that may increase the clarity and confidence of this Scoping Review’s message:

Author’s Response and Changes: Thank you for your feedback

Comment: First, I encourage the authors to emphasize effects of specific drugs in this class, and to include precise details about dose-effect outcomes; as the pharmacology of these drugs is often biphasic -with ascending and descending limbs on behavioral, cognitive, and emotional endpoints.

Author’s Response and Changes: The following has been added to Introduction (section Background) – line 65:

3-4- methylenedioxymethamphetamine (MDMA) has been shown to cause an enhanced mood, emotional warmth, empathy and a willingness to discuss emotionally charged memories [9]. Lysergic acid diethylamide (LSD) causes a feeling of closeness to others, openness, and enhanced emotional empathy [10]. Studies reveal that psilocybin enhances feelings of connectedness, increases openness, and increases altruism [11]. N, N-dimethyltryptamine (DMT) is known to create distortions in auditory and time perception and feelings of ego dissolution [12]. Several psychedelic drugs also have dose-effect outcomes, with high doses (≥ 3mg/kg) of MDMA producing antidepressant effects, some addictive potential and amnesia of fear conditioning memory, and low doses (≤ 1 mg/kg) not producing any of these effects [13]. Similarly, 5 mcg of LSD has been demonstrated to be the minimal dose required for observable effects. However, 20 mcg produces the most apparent effects [14].

In addition, I have also added the following references for this change:

NIDA. What are the effects of MDMA? . 13 Apr 2021 [cited 26 May 2024]. Available: https://nida.nih.gov/publications/research-reports/mdma-ecstasy-abuse/what-are-effects-mdma

Liechti ME. Modern Clinical Research on LSD. Neuropsychopharmacol. 2017;42: 2114–2127. doi:10.1038/npp.2017.86

Lowe H, Toyang N, Steele B, Valentine H, Grant J, Ali A, et al. The Therapeutic Potential of Psilocybin. Molecules. 2021;26: 2948. doi:10.3390/molecules26102948

Reckweg JT, Uthaug MV, Szabo A, Davis AK, Lancelotta R, Mason NL, et al. The clinical pharmacology and potential therapeutic applications of 5‐methoxy‐N,N‐dimethyltryptamine (5‐MeO‐DMT). Journal of Neurochemistry. 2022;162: 128–146. doi:10.1111/jnc.15587

Pantoni MM, Kim JL, Van Alstyne KR, Anagnostaras SG. MDMA and memory, addiction, and depression: dose-effect analysis. Psychopharmacology. 2022;239: 935–949. doi:10.1007/s00213-022-06086-9

Hutten NRPW, Mason NL, Dolder PC, Theunissen EL, Holze F, Liechti ME, et al. Mood and cognition after administration of low LSD doses in healthy volunteers: A placebo controlled dose-effect finding study. European Neuropsychopharmacology. 2020;41: 81–91. doi:10.1016/j.euroneuro.2020.10.002

Comment: Second, it will be useful to be sure an overview (perhaps additional Heading) of how any risks of bias are clearly avoided in the interpretation & consensus of meaning, as interpreted from study results. For example, I suspect highlighting discrepancies resulting from dose-related differences may bolster awareness of attention to crucial details in such a way as to avoid bias.

Author’s Response and Changes: The following has been added to Methods (section Data summarization and presentation) – line 190:

At this stage, risks of bias will be avoided by highlighting discrepancies that result from population, administration, or dose-related differences.

Comment: Finally, even though the Methods - under the Heading: Screening Strategy - mentions the use of two reviewers to verify inclusion and exclusion criteria, and even suggests a third, if necessary, I could argue there is utility in again identifying the use of two (or three reviewers) within the flow chart provided in the flow of Figure 1 (S1 Checklist).

Author’s Response and Changes: The following texts have been removed from Figure 1:

Records screened (n= )

Full-text articles assessed for eligibility (n= )

In their place, the following has been added:

Records screened by two reviewers (n= )

Full-text articles assessed for eligibility by two reviewers (n= )

---

## [Editor Report · Decision Letter 1]

7 Jul 2024

Investigating the impact of psychedelic drugs on social cognition defects: A scoping review protocol

PONE-D-24-10240R1

Dear Dr. Zhang,

We’re pleased to inform you that your manuscript has been judged scientifically suitable for publication and will be formally accepted for publication once it meets all outstanding technical requirements.

Kind regards,

Herb Covington, Ph.D.

Academic Editor

PLOS ONE
---

## [Editor Report · Acceptance letter]

19 Jul 2024

PONE-D-24-10240R1 

PLOS ONE

Dear Dr. Zhang, 

I'm pleased to inform you that your manuscript has been deemed suitable for publication in PLOS ONE. Congratulations! Your manuscript is now being handed over to our production team.

Kind regards, 

on behalf of

Dr. Herb Covington 

Academic Editor

PLOS ONE